# Characterization of Early and Late Damage in a Mouse Model of Pelvic Radiation Disease

**DOI:** 10.3390/ijms24108800

**Published:** 2023-05-15

**Authors:** Roberta Vitali, Francesca Palone, Ilaria De Stefano, Chiara Fiorente, Flavia Novelli, Emanuela Pasquali, Emiliano Fratini, Mirella Tanori, Simona Leonardi, Barbara Tanno, Eleonora Colantoni, Sara Soldi, Serena Galletti, Maria Grimaldi, Alessio Giuseppe Morganti, Lorenzo Fuccio, Simonetta Pazzaglia, Claudio Pioli, Mariateresa Mancuso, Loredana Vesci

**Affiliations:** 1Laboratory of Biomedical Technologies, Agenzia Nazionale per le Nuove Tecnologie, l’Energia e lo Sviluppo Economico Sostenibile (ENEA), 00123 Rome, Italy; 2AAT Advanced Analytical Technologies Srl, Via P. Majavacca 12, 29017 Fiorenzuola d’Arda (PC), Italy; 3Corporate R&D, Alfasigma S.p.A., Via Pontina km 30.400, 00071 Pomezia, Italy; 4Dipartimento di Scienze Mediche e Chirurgiche, Alma Mater Studiorum-Università di Bologna, Via Zamboni 33, 40126 Bologna, Italy

**Keywords:** mouse model, intestinal damage, ionizing radiation, intestinal stem cells, fecal markers

## Abstract

Pelvic radiation disease (PRD), a frequent side effect in patients with abdominal/pelvic cancers treated with radiotherapy, remains an unmet medical need. Currently available preclinical models have limited applications for the investigation of PRD pathogenesis and possible therapeutic strategies. In order to select the most effective irradiation protocol for PRD induction in mice, we evaluated the efficacy of three different locally and fractionated X-ray exposures. Using the selected protocol (10 Gy/day × 4 days), we assessed PRD through tissue (number and length of colon crypts) and molecular (expression of genes involved in oxidative stress, cell damage, inflammation, and stem cell markers) analyses at short (3 h or 3 days after X-ray) and long (38 days after X-rays) post-irradiation times. The results show that a primary damage response in term of apoptosis, inflammation, and surrogate markers of oxidative stress was found, thus determining a consequent impairment of cell crypts differentiation and proliferation as well as a local inflammation and a bacterial translocation to mesenteric lymph nodes after several weeks post-irradiation. Changes were also found in microbiota composition, particularly in the relative abundance of dominant phyla, related families, and in alpha diversity indices, as an indication of dysbiotic conditions induced by irradiation. Fecal markers of intestinal inflammation, measured during the experimental timeline, identified lactoferrin, along with elastase, as useful non-invasive tools to monitor disease progression. Thus, our preclinical model may be useful to develop new therapeutic strategies for PRD treatment.

## 1. Introduction

Pelvic radiation disease (PRD) is one of the most feared complications for patients suffering from abdominal or pelvic malignancies, i.e., colon, rectum, anus, prostate, testes, bladder, cervix, and womb [1]. Most of treatment regimens for pelvic cancers involve daily irradiation over the course of one or several weeks, achieving a cumulative dose of 40–50 Gy [2]. About 60–80% of patients subjected to radiotherapy develop acute or progress to chronic radiation enteritis; these are differentiated by the onset and clinical symptoms, causing, in some cases, therapy suspension [3]. Intestinal injuries such as fistulas, strictures, malabsorption, abdominal pain, and dysentery are just some clinical manifestations of the radiotherapy side effects, resulting in increasing hospitalization costs and irreversibly degraded quality of life [4,5]. The impact on human health of PRD has long been overlooked, but, in the era of cancer survivorship, the development of experimental models useful to increase the knowledge of the molecular mechanisms driving the onset of PRD is currently an emerging research area.

Being characterized by a rapid cell turnover, the intestine is very sensitive to radiation-induced damage, and this strongly limits the dose of radiation used for therapy. Following irradiation, the intestinal stem cells (ISCs) undergo massive apoptosis and show a cycle arrest in the G2/M phase [6]. However, the intestinal epithelium shows a good ability to recover after suffering damage.

The gut-on-a-chip system is emerging as an in vitro platform to improve radiobiological studies in terms of risk assessment, anti-tumor, and radio-protective drug discovery [7,8]. Furthermore, recent studies demonstrate that intestinal organoids can recapitulate physiological stress responses induced by ionizing radiation [9,10].

While these in vitro systems show promising benefits, they still have important limitations. In this setting, pre-clinical models of PRD allow for the evaluation of the long-term and systemic response to X-ray exposure, including survival, macroscopic manifestation (weight loss), dysbiosis, connections between inflammation and gut wall impairment, and the identification of potential biomarkers for clinical application. Furthermore, they are also pivotal to optimize the dose level and dose fractionation. Due to technical difficulties in replicating the clinical therapeutic procedures in animal models, i.e., treating a circumscribed area (distal section of the murine colon) with repeated exposures (fractionation), most studies are derived from mice exposed to total body (TBI) or whole abdomen irradiation (WAI) [11,12,13,14,15]. The literature data reveal that non-lethal intestinal damage can be induced by irradiation with 10 Gy WAI, while lethal intestinal damage can be triggered by irradiation doses ≥ 6 Gy TBI and >15 Gy WAI [16]. Notably, the high mortality rate together with the medullar and gonad toxicity in these preclinical experiments precluded clinically relevant investigations on the long-term side effects of pelvic irradiation. Indeed, a very limited number of studies evaluating long-term consequences of radiation-induced intestinal damage, following localized irradiation, are reported [17,18].

Herein, for the first time, we made an effort to better characterize a murine model of pelvic radiation disease by also using biomarkers easily translatable to human clinical studies. We have locally irradiated the mouse pelvic area with different daily dose fractions, shielding the remaining parts of the body with a lead-shield. None of the adopted dose fractionation regimens (i.e., 5 Gy for 5 days, 8 Gy for 4 days, and 10 Gy for 4 days) affected mouse survival, but we selected the 4 × 10 Gy irradiation protocol as the most effective for PRD induction on the basis of the highlights on tissue (number and length of colon crypts) and molecular (expression levels of genes involved in oxidative stress, cell damage, inflammation, and stem cell markers) changes in the colonic mucosa, respectively at long and short-term post irradiation. Long-term, these changes were associated with an altered differentiation, a decreased replicative/regenerative capacity, and a permanent degree of local inflammation, which at a macroscopic level resulted in a lack of body weight recovery; bacterial translocation from the gut to mesenteric lymph nodes was also increased and gut microbiota showed interesting variations compared to untreated animals. Importantly, in addition to elastase, we demonstrated that lactoferrin can be used as fecal marker to monitor the disease progression in our PRD mouse model. Altogether, our results pave the road for the development of new diagnostic, prognostic, and therapeutic strategies for the treatment of PRD, using this well characterized mouse model; for example, antagonists of inflammatory and oxidative stress targets to preserve ISCs and to prevent leaky gut could be taken into account as PRD treatment. Furthermore, the fecal markers analyzed here could be proposed to early predict PRD manifestations in oncological patients after a radiotherapeutic treatment.

## 2. Results

### 2.1. Evaluation of the Effects Induced by Different Irradiation Protocols

To compare the effects of our irradiation protocols (Figure 1A), different biological endpoints typically associated with the impairment in intestinal function were evaluated.

First, we monitored the mouse body weight three times a week. As shown in Figure 1B, during the irradiation sessions, a progressive decrease in the body weight was registered in all irradiated groups, reaching a reduction of about 6.5% compared to the average body weight of the sham-irradiated group at day 4 of X-ray exposure (sham vs. 5 × 5 Gy *p* < 0.01; sham vs. 4 × 8 Gy *p* < 0.05; sham vs. 4 × 10 Gy *p* < 0.05). Notably, three weeks after the last X-ray exposure, the weight of the 5 × 5 Gy group recovered, being comparable to the control group; afterwards, the weight trend curves of the two groups overlapped. On the contrary, even if the body weight of the 4 × 8 Gy and 4 × 10 Gy groups increased over time, the value never reached that of the sham group, remaining at approximately 95% until 38 days after the last X-ray exposure (sham vs. 4 × 8 Gy and 4 × 10 Gy *p* < 0.05) (Figure 1B).

Then, we carried out, by real-time PCR, the gene expression analysis of pro-inflammatory markers (*IL1-β*, *IL6* and *TNF-α*) in the colon tissue from mice sacrificed short-term (3 days) after the last X-ray exposure. The results show that in the 4 × 10 Gy group, 3 days after the last irradiation, there was still a significant increase in the IL6 expression (*p* < 0.01) (Figure 1C).

Finally, we carried out morphometric analyses of the intestinal epithelium at long-term (38 days) after the last X-ray exposure. The results showed that irradiation produced a significant decrease in the number of crypts/mm (*p* < 0.01) and in the crypt length (*p* < 0.05) only in the 4 × 10 Gy group compared to the controls (Figure 1D). Representative histological H&E images for each treatment performed are shown in Figure 1E.

Overall, our results indicated that the exposure at the dose of 10 Gy for 4 consecutive days (4 × 10 Gy) was the most effective treatment for radiation-induced colitis in terms of weight loss, molecular, and (long-term) morphological alterations of the colonic mucosa. Therefore, this irradiation schedule was selected to deeply investigate the early molecular and histological responses to irradiation damage and the residue damage after a long time.

### 2.2. Characterization of X-ray Effect in the Colon Tissue at Early Time

The gene expression of cytokines (*IL6*, *IL1-β*, *TNF-α*), genes involved in oxidative stress (*iNOS*, *COX2*), and apoptosis (*BAX*, *Bcl2*), as well as of genes involved in the cell cycle (*p21*, *p19*) and tight junction components (*ZO-1*) were analyzed in the colon tissue of mice sacrificed 3 h after treatment with 4 × 10 Gy, by real-time PCR. The results showed statistically significant increased expression levels of *IL6* (*p* < 0.001), *iNOS* (*p* < 0.01), *BAX* (*p* < 0.0001), and *p21* (*p* < 0.0001) in irradiated animals compared to controls (Figure 2A–D). No changes were shown for the other genes evaluated (Appendix A). To assess whether the increased gene expression observed 3 h after irradiation persists or reaches the normalization, an analysis of *IL6*, *iNOS*, *BAX*, and *p21* mRNA levels was performed also at 3 days after the last X-ray exposure. The results showed that *IL6* (*p* < 0.05), *p21* (*p* < 0.01) and *BAX* (*p* < 0.0001) expression levels remain significantly increased in the group treated with 4 × 10 Gy compared to controls (Figure 2F,H,I).

In order to characterize the apoptotic damage caused by irradiation, the number of apoptotic cells/crypt was determined 3 h and 3 days after the last exposure. The results showed a statistically significant increase (*p* < 0.0001) in the number of apoptotic cells per crypt in the irradiated group compared to controls 3 h after the last irradiation (Figure 2E). This parameter reaches the physiological value 3 days after the last exposure with 4 × 10 Gy (Figure 2J).

Finally, the expression of the intestinal alkaline phosphatase (*ALPI*), a digestive enzyme that acts in the detoxification of lipopolysaccharide and in the prevention of bacterial translocation in the gut [19], was analyzed. We found that, at 3 days post-irradiation, *ALPI* expression was significantly decreased (*p* < 0.0001) compared to controls (Figure 2K).

### 2.3. Gene Expression Analysis of Intestinal Stemness Markers following Irradiation

The intestinal epithelium is organized into crypt and villus regions, with the stem and progenitor zone localized in the crypt. Tissue regeneration in the gut is mediated by ISCs that reside at the base of the intestinal crypts, and include columnar cells at the base of the crypt (CBC) and cells in the +4 position counting from the bottom of the crypt (+ 4 cells). The progeny of ISCs, called transient amplification cells (TAs), expands through several cycles of mitosis as it migrates upward along the crypt axis. Close to the intestinal lumen, TA cells undergo cell cycle arrest and terminal differentiation. In order to understand the impact of radiation on the intestinal epithelium regeneration process, we analyzed specific markers for each intestinal stem cell population 3 h and 3 days after the last irradiation with 4 × 10 Gy, such as Leu-rich repeats containing G protein-coupled receptor 5 (Lgr5), achaete-scute family bHLH transcription factor 2 (Ascl2), and SPARC related modular calcium binding 2 (Smoc2): CBC markers; Leu-rich repeats and immunoglobulin-like domains 1 (Lrig1), Telomerase reverse transcriptase (Tert), and Polycomb complex protein BMI-1 (Bmi1): +4 cell markers; Prominin-1 (Prom1), Ephrin receptor B2 (Ephb2), and Zinc finger protein 277 (ZNF277): TA cell markers.

Treatment with 4 × 10 Gy induced a significant decrease in the expression of *Lgr5* (*p* < 0.001), *Smoc2* (*p* < 0.01), and *Lrig1* (*p* < 0.001) at 3 h after irradiation (Figure 3A,B) and this persisted after 3 days, only for the *Lgr5* (*p* < 0.05) and *Lrig1* (*p* < 0.05) genes (Figure 4A,B).

### 2.4. Evaluation of the Effects Induced by Irradiation on Colonic Mucosa at Long Time

We next carried out extensive morphometric, histochemical, and molecular analyses on colonic mucosa to characterize the damages at a long time after radiation exposure. In detail, the length of the crypts, the number of Goblet cells, the proliferative index (Ki67-positive cells), the inflammation status (Iba1-positive cells), vascular density and vessel area, and the expression of *TNF-α* were evaluated 38 days after the last irradiation. The results showed that there was a significant decrease in the length of the crypts (*p* < 0.001), the Goblet cell number (*p* < 0.01), and the number of Ki67+ cells (*p* < 0.0001). Furthermore, we detected a significant increase in Iba1+ cells (*p* < 0.05) and in *TNF-α* expression in the colonic mucosa after exposure to 4 × 10 Gy compared to controls (Figure 5 A–E). A significant positive correlation between the number of Ki67+ cells and crypt length was observed, both in the sham and in the X-ray-treated group (Figure 5F). No change in the vascular area was found. Conversely, a significant increase in vascular density was detected after X-Ray treatment (Figure 5G,H). Finally, the expression of the intestinal stemness markers, *Lgr5* and *Lrig1*, that were altered up to three days post-irradiation, returned to the basal level 38 days post-irradiation (Figure 5I).

### 2.5. Time-Course Analysis of the Fecal Elastase and Lactoferrin after 4 × 10 Gy Irradiation

Fecal samples from each mouse were collected 1, 3, 10, 25, and 38 days after the last X-ray exposure, and were analyzed, by ELISA assay, for the content of two known fecal markers of intestinal inflammation, neutrophilic elastase and lactoferrin. The results showed that starting from the first day following the last irradiation, a significant increase in elastase (*p* < 0.0001) and lactoferrin (*p* < 0.0001) content was observed in the irradiated compared to the sham group. Increased levels of both fecal markers were maintained until 38 days after treatment (elastase: 1, 3, and 10 days *p* < 0.0001, 25 and 38 days *p* < 0.01; lactoferrin: 1, 3, and 10 days *p* < 0.0001, 25 and 38 days *p* < 0.001) (Figure 6A,B).

### 2.6. Evaluation of Bacterial Translocation

As reported in Figure 7, the comparison was run for the two time points (3 vs. 38 days) on sham and irradiated mice in order to verify the differences in bacterial translocation due to the acute enteritis phase, or delayed damage and concomitant possible increase in bacterial presence in different districts. Bacterial translocation to a physically more distant organ (liver) showed similar trends in the two time points, without any statistically significant difference between groups (Figure 7A for liver). On the other hand, the model confirmed its validity, in particular representing the damage in mesenteric lymph nodes, which were sampled as the first point in which bacterial translocation could occur. In mesenteric lymph nodes, both time points clearly indicated the increased bacterial presence in irradiated mice compared to controls, and a statistical significance was reached at a time point of 38 days (Figure 7B).

### 2.7. Evaluation of Microbiota Fluctuations

The sequencing raw reads were subjected to a series of quality control steps for the removal of PCR and sequencing artefacts, and this resulted in a reduction in the sequence dataset by ~45.84%.

The remaining sequences showed an achieved coverage of 99.3 ± 0.6% of the existing bacterial diversity among OTUs of ≥0.1% in relative abundance, according to the Good’s coverage estimate.

The within-sample diversity (aka α-diversity) of the samples was assessed using four associated indices, indicative of the diversity of the various levels of dominance (Shannon index, inverse Simpson, and Fisher’s α index), while the remaining shows the predicted population richness (ACE) (Figure 8). The overall results showed a decrease in alpha-diversity indices measured in irradiated mice compared to sham animals. In particular, the reduction in alpha diversity values appeared at Day 3 and was confirmed with higher impact at Day 38. This was evident for inverse Shannon and Simpson indices, also reaching statistical significance in reduction at Day 38, while, for Fisher’s index, Day 3 showed values similar to those measured in non-irradiated mice but the difference was increased at Day 38, confirming the ability of PRD to result in long-term effects. The ACE index did not present any variation in the timepoints.

The dominant phyla were investigated, and Bacteroidetes, Firmicutes, and Verrucomicrobia were identified as the most represented, followed by Deferribacteres, Proteobacteria, and Tenericutes. The behavior of the samples confirmed the trend identified in alpha-diversity indices, with irradiation effects shown at Day 3 and further reinforced at Day 38. In detail, Bacteroidetes were strongly increased by X-ray application at Day 3, as shown in Figure 9, and their amount was even higher at Day 38; on the opposite side, Firmicutes were reduced by irradiation at Day 3 and the difference became statistically significant at Day 38 compared to non-irradiated animals. Finally, Verrucomicrobia did not achieve statistical significance but showed a clear behavior in increasing the relative abundance from Day 3 to Day 38 compared to untreated animals. Other phyla were represented in low relative abundance, as is shown in Appendix A.

Interesting results were provided by the identification of the most represented bacterial families within the samples (Figure 10). Ten families were found to be mostly abundant, and their variations supported the macroscopic variations that occurred in dominant phyla. Lachnospiraceae, Ruminococcaceae, and Marinifilaceae showed a statistically significant reduction in irradiated animals at both timepoints compared to untreated animals, while Prevotellaceae, Lactobacillaceae, and Rikenellaceae were increased or reduced, respectively, after irradiation with a statistically significant difference at Day 3, and with a reduction in the differences at Day 38. These results further supported the ability of X-radiation to induce long-term progressive modifications in microbiota.

Other families were also modified by radiation, without reaching statistical significance, such as Erysipelotrichaceae and Akkermansiaceae.

## 3. Discussion

The intestine is one of the most radiosensitive organs in the body. Exposure to radiation for the clinical treatment of cancers in the abdominal and pelvic cavity may lead to acute and/or chronic intestinal injury, which significantly reduces the quality of life and also adds an extra economic burden to the health care system.

A deeper understanding of the radiation-induced intestinal pathogenesis is crucial to prevent or mitigate complications after radiation therapy in the pelvic area. In the present study, we used a mouse model of radiation-induced enteritis with the aim of studying the short- and long-term consequences and the molecular mechanisms involved in response to irradiation.

We compared three irradiation protocols to identify the one best recapitulating the PRD pathology observed in patients undergoing radiotherapy. To ensure mouse survival, the irradiation field was limited, by shield, to a 1 cm^2^ area and the total dose was fractionated. The 4 × 10 Gy irradiation protocol was the most effective to induce enteritis in terms of percentage of weight loss, histological damage, and alteration in gene expression. Although we adopted a different schedule for X-ray exposure, consisting of fractions every 24 h instead of every 12 h, making our protocol easier to perform, our results are in agreement with those obtained by Bull and co-workers [17]. Consequently, this irradiation schedule was selected to characterize the early molecular response to radiation damage. However, we must admit that the irradiation regimen we used is not entirely comparable to those prescribed in clinical routine (in stereotactic treatments), since the latter are often delivered every other day. Nevertheless, our choice originated from the need to select a treatment that is not lethal, but at the same time not too well tolerated, in order to deepen our knowledge of the radiobiological mechanisms of the radiation-induced damage to the bowel.

In our study, animals were sacrificed 3 h after the last exposure. According to the literature, irradiation induced an increase in the number of apoptotic cells [6], expression of pro-inflammatory cytokines [20], oxidative stress [21], cycle blocking, and apoptotic response. Although apoptosis and oxidative stress recovered shortly after irradiation, cell cycle arrest and inflammation persisted at 38 days post-irradiation.

The Intestinal epithelium can rapidly self-renew every 4–5 days, being the fastest tissue in mammals [22]. Current models favor the existence of two stem cell populations, CBC and the +4 stem cells, which are thought to be active and quiescent stem cells, respectively. Transit-amplifying (TA) progenitors arise from the stem cell compartment and differentiate into absorptive enterocytes or secretory goblet, enteroendocrine, tuft, or Paneth cells. Lgr5, Ascl2, and Smoc2 have been identified as selective markers of CBC [23,24]. Homeodomain-only (Hopx), Lrig1, and Tert are considered +4 stem cell population markers [25]. Finally, other studies have revealed the expression of the following markers in the TA cells: Prom1 [26], Musashi homologue-1 (Msi1) [27], Olfactomedin-4 (Olfm4), and Ephb2 [28].

Over the years, there has been increasing interest in better understanding the regenerative mechanisms of the intestine after chemotherapy or radiation treatment, given their relevance in the context of oncological therapies [29]. It has been reported that when the mouse rectum is exposed to radiation doses lower than 17.5 Gy, radiation damage can be entirely repaired through intestinal stem cell regeneration, but if the single dose exceeds 20 Gy, the ISCs are unable to regenerate and repair the damage [30]. Liang and co-workers found that in C57Bl/6 mice, Lgr5 expression was decreased 3.5 and 5 days after WAI with a single dose of 12 Gy, but recovered 10 days post-irradiation [31]. It has been hypothesized that +4 stem cells, since they are quiescent, have a greater ability to resist radiation-induced damage, and they could be at the forefront during the regenerative process. Consistently with this assumption, a decrease in Lgr5+ CBCs was reported following irradiation, supporting the hypothesis that radiation-induced Lgr5+ cell depletion causes the reactivation of previously quiescent +4 stem cells [32,33,34,35]. However, the alteration of intestinal stem cells in models in which the irradiation was fractionated and localized has not been analyzed so far. Our data show that at an early time, the intestinal cell compartments affected by irradiation are CBC and +4 cells. While the depletion of CBC is supported by the literature, the decreased *Lrig1* expression, indicative of a decrease in +4 cells, was not previously reported and this population was considered as radioresistant [35]. This apparent discrepancy could be due to the different schedule adopted for X-ray exposure, and at the timing during which the analysis was performed. Finally, consistently with their radio-resistance, we did not observe any depletion of TA cells [36].

Since perturbations of the stem cell compartments can result in alterations in the intestinal epithelium, which may not be able to be correctly renewed, we histologically characterized the damage in the colon through extensive morphometric and immunohistochemical analyses at long post-irradiation time. We observed a significant decrease in crypt length, number of Goblet and Ki67+ cells, and a prolonged inflammation of the colonic mucosa.

Overall, our results suggest that the impairment in Goblet cells observed in our model is associated with the altered differentiation of damaged ISCs, highlighted shortly after irradiation.

As previously stated, pelvic irradiation induces a rapid inflammatory response followed by a functional impairment of the intestinal barrier, delayed in the colon, with respect to the ileum. Furthermore, epithelial barrier damage promotes bacterial translocation into mesenteric lymph nodes [37]. We demonstrated that irradiation induces an increased bacterial presence in mesenteric lymph nodes, persisting a long time after irradiation and thus causing the development of chronic effects.

The same chronic effects were found in the variations that occurred in microbiota composition, as documented by several authors about the fluctuations identified after irradiation, and in both initial and long-term stages of associated enteritis. The decrease in Firmicutes and the concomitant increase in Bacteroidetes phyla [38,39], as well as the reduction in families belonging to short-chain fatty acid producers (Ruminococcaceae and Lachnospiraceae) was documented in animals and humans [40,41,42,43] after X-ray administration. Our results found further support in other papers focused on dysbiosis and radiation-induced enteritis, with the variations in Rikenellaceae representatives in different gut districts [40] and the increase in genera belonging to the Prevotellaceae family [44] in patients with radiation-enteritis, both in the acute and chronic phases of the process, as confirmed by the reduction in Marinifilaceae members, recently reported as decreased in inflammatory bowel disease (IBD) patients [45].

Non-invasive inflammatory biomarkers have been proposed to monitor the degree of radiation-induced gut-wall injury in the distal bowel and could be useful for the management of the disease. Among these, elastase [46] in mice, and lactoferrin in humans and in rats [47,48,49] were identified as markers. As a limitation, it has been reported that fecal elastase is strongly influenced by the surrounding environment [46].

In our study, a time-course analysis of these fecal markers shows a significant increase in their levels starting from the day following the last irradiation and this increase is maintained up to 38 days after treatment, suggesting that in agreement with what has been observed in patients, a damage capable of inducing sub-clinical inflammation persists even long after irradiation.

According to previous studies, we set up our PRD model on male mice as our shielding systems could ensure the protection of the testis from exposure to radiation, thus preventing indirect hormonal effects. Although no differences in radiation-induced intestinal side effects were reported between male and female mice in pre-clinical research [50], sex-related differences in the efficacy of therapeutic strategies associated with dimorphism in gut microbiota were described [1]. Another factor that could affect the response to irradiation in PRD is the age of the patients, considering changes in inflammatory and immune responses occurring during ageing. We are therefore planning to extend our study to female as well as aged mice in future experiments to explore the role of sex- and/age-associated effects in PRD.

Patients with a healthy gut flora before radiation therapy have lower post-irradiation side-effects [51], soliciting the identification of non-invasive and reliable markers to predict patients that will develop radiation-induced gastrointestinal disease. In this setting, the fecal biomarkers analyzed may be proposed in patients for PRD diagnosis or prognosis during radiotherapy, and to help the clinician to modulate radiotherapy, as well as to monitor the effects of radio-protective drugs or natural supplements proposed to improve radio-induced enteritis.

Overall, our results indicate that the identified irradiation protocol causes a damage associated initially with oxidative stress, inflammation, and a high degree of apoptosis, typical of the response to X-rays, and a decrease in CBC and +4 stem cells. This damage, in the long-term, induces perturbations of the intestinal mucosa associated with an altered differentiation, a decreased replicative/regenerative capacity, and a permanent degree of local inflammation, which at a macroscopic level manifests in the impossibility to completely recover the body weight loss.

## 4. Materials and Methods

### 4.1. Animals

Forty nine fifty-five days old male C57Bl/6 mice were purchased from Charles River Laboratories (Lodi, Italy) and housed one per cage. Treatments started after two weeks of acclimatization in our animal facility, when mice were 63–69 days old. All animals were housed at a constant temperature (20 °C) with a 12 h/12 h dark/light cycle; food and water were available ad libitum. All experiments involving animal studies were performed according to the European Community Council Directive 2010/63/EU, approved by the local Ethical Committee for Animal Experiments of the ENEA, and authorized by the Italian Ministry of Health (n° 839-2020-PR; n° 739/2021-PR).

### 4.2. Irradiation Procedures

Mice were weighted and randomized according to the body weight in 2 groups: sham and RX. Before irradiation, animals were anesthetized with sodium pentobarbital (40 mg/kg) and then locally exposed to X-rays using a Gilardoni CHF 320G X-ray generator (Gilardoni S.p.A., Lecco, Italy) operated at 250 kVp, 15 mA, with filters of 2.0 mm of Al and 0.5 mm of Cu. The dose rate was 0.89 Gy/min at an irradiation distance of 67.7 cm. The irradiated pelvic area was 1 cm^2^, whilst the remaining parts of the body were shielded with a lead-shield. Mice were exposed to three different irradiation protocols, as shown in Figure 1A, i.e., 5 Gy for 5 days, each 24 h apart, for a total of 25 Gy; 8Gy for 4 days, each 24 h apart, for a total of 32 Gy; 10 Gy for 4 days, each 24 h apart, for a total of 40 Gy. Sham-irradiated mice were subjected to all procedures aforementioned, except for X-ray delivery. The number of mice per group and time points are summarized in Figure 1A.

In the second set of experiments, mice were weighted and randomized according to body weight in 2 groups: sham and 4 × 10 Gy and sacrificed at three distinct time points: 3 h (*n* = 9), 3 days (*n* = 9), and 38 days (*n* = 12) after last X-ray exposure (Appendix A). To evaluate the presence of local inflammation, fecal samples were collected at 1, 3, 10, 25, and 38 days after X-ray exposure and stored at −80 °C.

During experiments, animals were monitored every two days for changes in weight and appearance. At the sacrifice, colon-rectal tissue was excised, from the anus and proximally, rinsed in PBS, and quickly frozen on dry ice or fixed in 10% formalin for further molecular or histological analysis, respectively.

### 4.3. Morphometric Analyses of Colon-Rectum

The distal colon (1 cm colon-rectum) was collected from mice sacrificed at 3 and 38 days after RX exposure and from the age-matched sham-irradiated group. Samples were fixed in 10% buffered formalin and embedded in paraffin wax according to standard procedures. All analyses were carried out using a double-blind method.

#### 4.3.1. Morphometric Measurements

After hematoxylin and eosin staining (H&E), morphometric measurements (number, and length of crypts) were performed on three colon sections of 4 µm collected at 100 µm intervals, using the software NIS-Elements BR version 5.30.10 (Nikon Instruments Europe B.V., Florence, Italy). The number of the crypts per mm was calculated from 2 frames per each colon sample. Crypts were considered well-oriented and counted only if the crypt–villus junctions on both sides of the crypt were intact. The measurement of crypt length was performed on a total of 15 well-oriented crypts for each colon sample.

#### 4.3.2. Assessment of Vascular Function

On H&E transverse colon sections of sham and X-ray-irradiated animals, 38 days after the last irradiation, their vascular density (number of vessels for total circumference) and area (µm^2^) were analyzed. Two sections (20× magnification) for each animal were evaluated.

#### 4.3.3. Assessment of Goblet Cell Number

For goblet cell staining, sections were stained with Alcian Blue solution pH 2.5 (Sigma-Aldrich, St. Louis, MO, USA) following standard procedures. The number of goblet cells per crypt was quantified using the software NIS-Elements BR (Nikon Instruments Europe B.V.). Three sections for each animal, and 3020 crypts/section, were analyzed. To be included in the quantification, crypts had to be open toward the lumen and stretch at least two-thirds of the distance between the lumen and muscularis mucosae.

#### 4.3.4. Quantification of Apoptosis

The quantification of apoptosis was performed on H&E-stained colon tissue sections by counting cells showing signs of nuclear chromatin condensation. The number of apoptotic cells was evaluated on 30 crypts/section for the colon sample.

### 4.4. Immunohistochemistry

Sections (4 µm) of paraffin-embedded intestinal colonic samples per each mouse were prepared following the standard protocol. Briefly, dewaxing and antigen retrieval procedures were performed by microwave oven heating using Antigen Retrieval Citrate Buffer pH 6.0 (ab93678) (Abcam, Cambridge, UK). Afterward, sections were washed in water for 5 min and peroxidases was inhibited by incubation in 3% H_2_O_2_ for 10 min. Sections were treated with 5% bovine serum albumin (Santa Cruz Biotechnology, Santa Cruz, CA, USA) for 30 min and incubated with primary anti-Ki67 (IHC-00375) (Bethyl, Montgomery, TX, USA; 1:500) and anti-Iba1 (Abcam, 1:2000) antibodies o.n., at 4 °C in a moist chamber. Sections were incubated with secondary antibody (goat anti-rabbit biotinylated 1:200) (Vector Laboratories, Burligame, CA, USA). Then, the slides were developed with Vectastain^®^ Elite^®^ ABC Kit peroxidase and were detected with Vector Novared substrate kit (VectorLaboratories), as suggested by the providers, to visualize the antigen. Finally, samples were counterstained with hematoxylin.

The proliferative index was calculated using NIS-Elements BR (Nikon Instruments Europe B.V.) software as the number of Ki67-positive cells per crypt, with 15 crypts for each animal.

A quantitative analysis of Iba1 levels on colon sections (4 frames for each animal) was performed using the software HistoQuest^®^ 6.0 (TissueGnostic, Vienna, Austria). Iba1 levels were expressed as sum intensity of Iba1+ area/μm2.

### 4.5. RNA Isolation and Real-Time PCR Analysis

A fragment of colon tissue was collected from each mouse and immediately snap-frozen in liquid nitrogen. Total RNA was isolated from each sample using the mini RNeasy kit (QiaGen GmbH, Hilden, Germany), and 1µg of total RNA was reverse transcribed by IScriptTM cDNA Synthesis Kit (BioRad, Hercules, CA, USA). Real-time PCR was carried out by a BioRad CFX96 TouchTM Real-Time PCR Detection System using SsoAdvanced Universal SYBR Green super Mix (BioRad). The expression level of each mRNA was assessed using the ΔΔCT method, and Gadph was used as housekeeping gene for normalization.

All primers used in this study are summarized in Appendix A.

### 4.6. Fecal Extraction

Murine stool specimens were collected and stored at −80 °C. Each sample was weighted and dissolved in denaturing buffer 1× (ScheBo Biotech AG, Giessen, Germany) to obtain a final concentration of 100 mg/mL of feces. Samples were vortexed for 1 min, placed in orbital shaking for 1 h at room temperature, and then were centrifuged for 10 min at 12,000 r.p.m. at 4 °C, twice. The supernatants (fecal extracts) were collected and stored at −80 °C.

### 4.7. Enzyme-Linked Immunosorbent Assay (ELISA)

ELISA Mouse Neutrophil Elastase/ELA2/Elane Pikokine kit (Boster Bio, Pleasanton, CA, USA) was used for the quantification of murine elastase concentration in fecal samples (diluted 1:2), according to the manufacturer’s instructions. Fecal lactoferrin levels in murine samples (diluted 1:10) were determined by ELISA Mouse LF/LTF/Lactoferrin kit (LSBio, Seattle, WA, USA). Positive signals were measured spectrophotometrically at a wavelength of 450 nm by GloMax^®^ Explorer Multimode Microplate Reader (Promega Italia S.r.l., Milan, Italy). The concentration of fecal elastase and lactoferrin level was expressed as pg/mL.

### 4.8. Quantification of Bacterial Translocation and Microbiota Profiling

Real-time PCR analysis was performed on DNA extracted from tissue samples collected with the aim to quantify the bacterial load due to translocation from the gut to organs (mesenteric lymph nodes and liver). Tissues were pre-treated with a Dispase II solution (Roche Diagnostics, GmbH, Mannheim, Germany) for a preliminary disaggregation (2.4 U/mL), and then bacterial DNA was extracted using the FastDNA SPIN Kit for Soil and FastPrep Instrument (MP Biomedicals, Santa Ana, CA, USA). The extracted DNA was quantified using the PicoGreen method of the Quant-iT™ HS ds-DNA assay kit in a Qubit™ fluorometer (Invitrogen, Carlsbad, CA, USA) and verified. All samples were diluted at the proper concentration suited for the real-time PCR reaction.

All real-time PCR reactions were run in StepOnePlus Real-Time PCR (Thermo Fisher Scientific, Waltham, MA, USA), and SsoAdvanced Universal SYBR Green Supermix (Bio-Rad) was used to correlate the amount of PCR product with the fluorescence signal. Bacterial DNA, extracted from tissues, was diluted to the proper concentration (2.5 ng/µL) and amplified with universal primers in order to quantify total bacteria [52].

The amount of the target was finally calculated using standard curves derived from known concentrations of genomic DNA certified by the International Collection DSMZ.

After the amplifications, the raw data obtained were adjusted on the basis of the appropriate dilution factor, with the aim to refer each result to the bacterial load present in 1 g of tissue.

DNA extracted from fecal samples was also used for next generation sequencing (NGS) analyses of microbiota composition. The V4-5 hypervariable regions of the bacterial 16S rRNA gene were amplified and sequenced by the Integrated Microbiome Resource Institute (Dalhousie University, Halifax, Nova Scotia, Canada). Amplicon libraries were generated with primers based on the 515FB (5′-GTGYCAGCMGCCGCGGTAA-3′)/926R (5′-CCGYCAATTYMTTTRAGTTT-3′), as suggested previously [53]. The sequencing instrumentation, methodology, and chemistry were based on the Illumina MiSeq instrument using the 2 × 300 bp paired-end v3 chemistry as detailed by Comeau [54].

### 4.9. Sequence Data Preparation and Statistics

The sequences were quality trimmed with Trimmomatic v0.39 [55] using the default settings (with the exception if 5′ trimming) and a minimum length cutoff of 100 bp after the sample index trimming (the average amplicon length was 410 bp). Total amplicon sequences were reconstructed through the assembly of the read pairs with the FLASH v1.2.11 [56] software using the default parameters. Further, sequence screening for sequencing errors and PCR-introduced chimeras, the alignment to reference databases, and generation of OTU matrices were performed with the Mothur v1.45.2 [57]. The sequences were then aligned against the Silva v138 16S rRNA gene reference alignment database, and sequences failing to align were considered non-specific and were removed from downstream analysis. Chimeric amplicons were identified and removed using the abundance-based de novo UCHIME v4.2.40 [58] approach. Sequence distances were calculated for the remaining aligned sequences, while the clustering of sequences into 0.03 distance-defined operational taxonomic units (OTUs) was performed with USEARCH [59].

Sequencing effort coverage and α-diversity indices were calculated with the entropart v1.6.6 [60] and the vegan v2.5.7 [61] packages of the R software [62]. Shannon index, the reciprocal Simpson index, and the Fisher’s α index were calculated as measures of the alpha-diversity of the samples.

The non-parametric Kruskal–Wallis test, followed by the Wilcoxon rank sum test (if the Kruskal–Wallis test was significant), was used for identifying differentially abundant OTUs among the different treatments and within the same group and timepoint in order to assess the potential effect at an OTU level, as previously suggested [63].

### 4.10. Statistics

Statistical analysis for significance was determined using GraphPad InStat software 8. Data were presented as mean ± SD or SEM. The Kolmogorov–Smirnov test was used to assess whether data were sampled from populations following the Gaussian distribution. The comparison between groups was performed using a Mann–Whitney U test or *t*-test. The correlation between Ki67+-cells/crypt and crypt length values in the sham and 4 × 10 Gy group was assessed with the Spearman rank correlation test. Differences were noted as significant * *p* ≤ 0.05, ** *p* ≤ 0.01, *** *p* ≤ 0.001, **** *p* ≤ 0.0001.

## 5. Conclusions

In conclusion, the proposed radio-induced enteritis mouse model, based on fractionated irradiation of a limited area, represents an improvement in the models already available in the literature, where a single dose is mainly used on the whole body or on the entire abdomen. Furthermore, to support the model proposed by Bull and collaborators [17], with a fractionated and localized dose, our work has also provided exhaustive molecular and histological analyses of the short- and long-term radiation consequences, thus clarifying the mechanisms involved in the pathogenesis of radio-induced enteritis. A further advantage of this model was the identification of the biomarker lactoferrin, beyond the elastase marker already known, as a fecal marker to monitor the persisting intestinal inflammation in oncological patients subjected to pelvic radiotherapy. Accordingly, it may be a suitable model to develop therapeutic strategies for the treatment of this pathology.

## Figures and Tables

**Figure 1 ijms-24-08800-f001:**
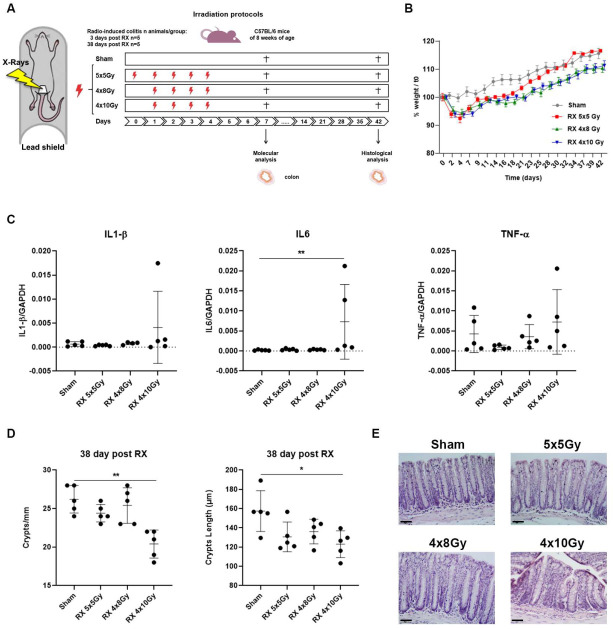
(**A**) Irradiation set up and experimental design. The irradiated pelvic area is 1 cm^2^. **┼** indicates day of sacrifice. (**B**) Percentage of weight loss during the three irradiation protocols of treatment, 4 days from the beginning of X-ray treatment (5 × 5 Gy *p* < 0.01; 4 × 8 Gy *p* < 0.05; 4 × 10 Gy *p* < 0.05 vs. sham), and 38 days after the last X-ray exposure (4 × 8 Gy and 4 × 10 Gy *p* < 0.05 vs. sham). (**C**) Gene expression analysis of the pro-inflammatory cytokines: *IL1-β*, *IL6*, and *TNF-α* at 3 days after the last X-ray exposure. (**D**) Histological analysis of the number and the crypts length (µm) on colonic mucosa of treated mice at 38 days after the last X-ray exposure. (**E**) Representative histological images for each treatment performed are shown in the right panel. Scale bar = 50 µm. *n* = 5 each experimental group. *p* < 0.05 (*), *p* < 0.01 (**).

**Figure 2 ijms-24-08800-f002:**
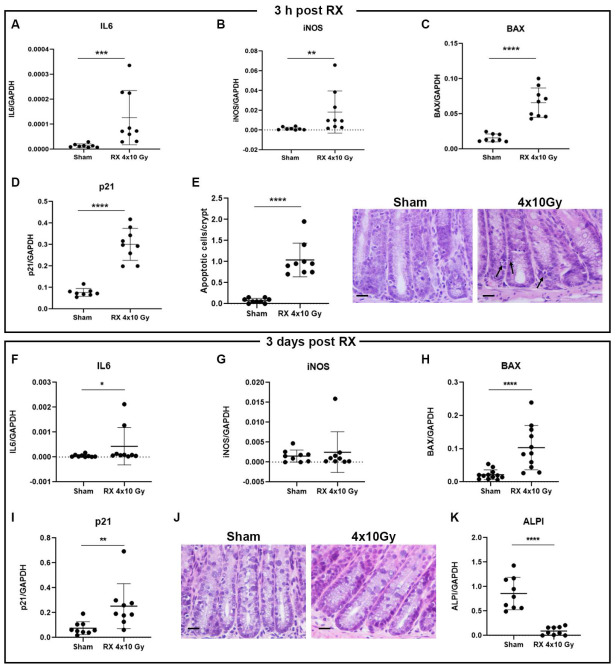
Gene expression analysis of *IL6*, *BAX*, *iNOS*, and *p21* in colonic mucosa of mice 3 h (upper panel **A**–**D**) and 3 days (bottom panel **F**–**I**) after the last X-ray exposure (4 × 10 Gy). Apoptotic cells 3 h (**E**) and 3 days (**J**) after the last X-ray exposure (4 × 10 Gy). (**K**) Gene expression analysis of *ALPI* 3 days after the last irradiation. Scale bar = 50 µm. *n* = 9 each experimental group. *p* < 0.05 (*), *p* < 0.01 (**), *p* < 0.001 (***), *p* < 0.0001 (****).

**Figure 3 ijms-24-08800-f003:**
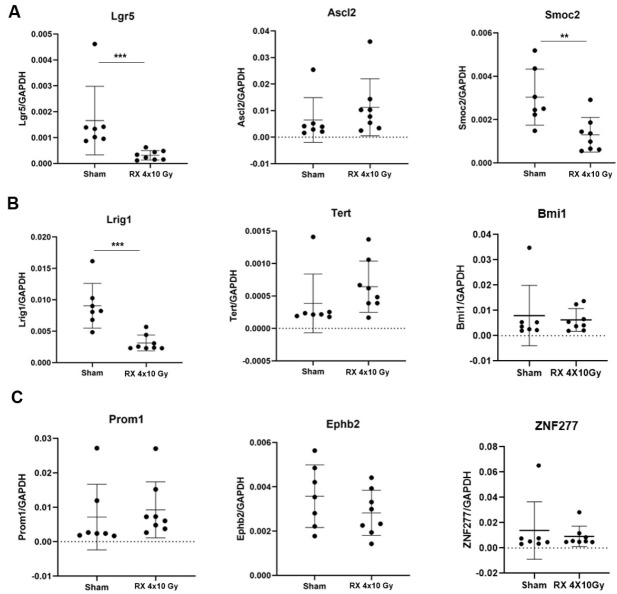
Gene expression analysis of specific markers for the following stem cell populations (**A**) CBC, (**B**) +4 cells, (**C**) TA cells, on colonic mucosa of mice 3 h after the last X-ray exposure (4 × 10 Gy). *n* = 9 each experimental group. *p* < 0.01 (**), *p* < 0.001 (***).

**Figure 4 ijms-24-08800-f004:**
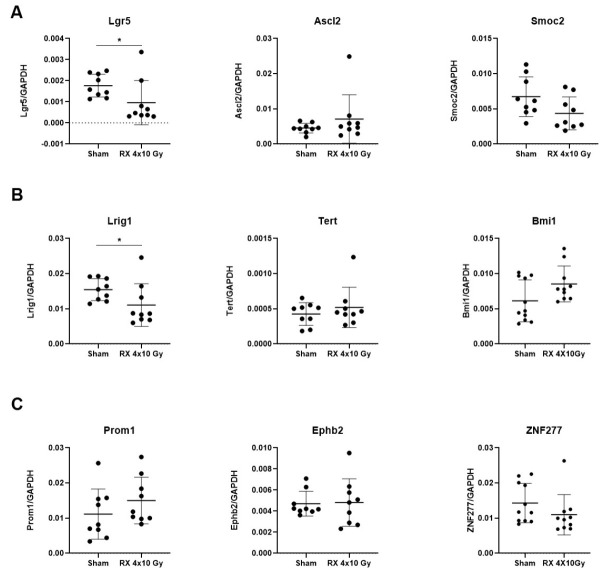
Gene expression analysis of specific markers for the following stem cell populations (**A**) CBC, (**B**) +4 cells, (**C**) TA cells, on colonic mucosa of mice 3 days after the last X-ray exposure (4 × 10 Gy). *n* = 9 each experimental group. *p* < 0.05 (*).

**Figure 5 ijms-24-08800-f005:**
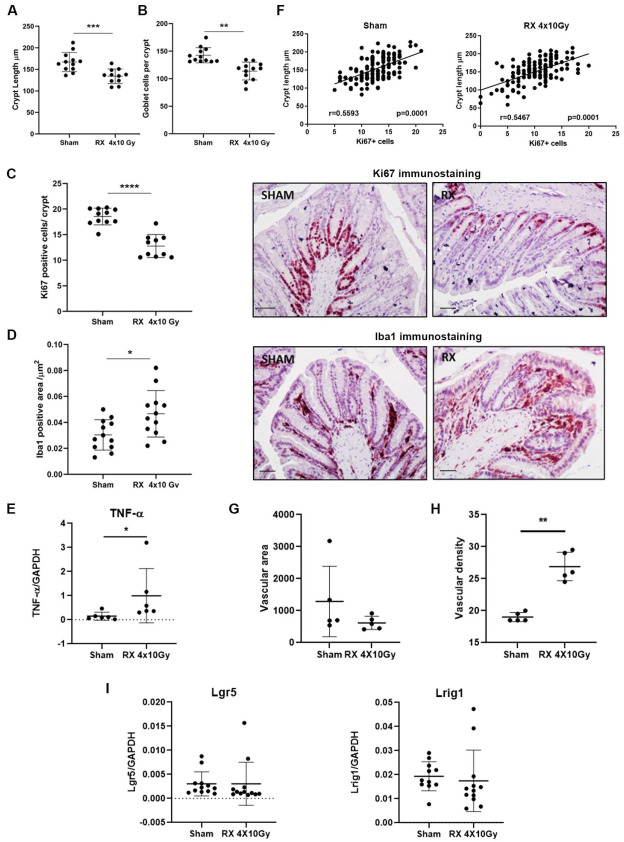
Histological, immunohistochemical, and molecular analysis of colonic mucosa of mice 38 days after the last X-ray exposure (4 × 10 Gy): (**A**) crypt length (µm), (**B**) number of Goblet cell/crypt, (**C**) number of cell Ki67+/crypt, and (**D**) Iba1 positive (area/µm^2^). Representative immunohistochemical images for each treatment are reported in the right panel. Scale bar = 50 µm. (**E**) Gene expression analysis of the pro-inflammatory cytokine *TNF-α*. (**F**) Spearman’s correlation analysis between crypt length and number of Ki67 positive cell in sham and in X-ray-treated group. (**G**) Vascular area. (**H**) Vascular density. (**I**) Gene expression analysis of stemness markers *Lgr5* and *Lrig1*. *n* = 12 each experimental group. *p* < 0.05 (*), *p*< 0.01 (**), *p* < 0.001 (***), *p* < 0.0001 (****).

**Figure 6 ijms-24-08800-f006:**
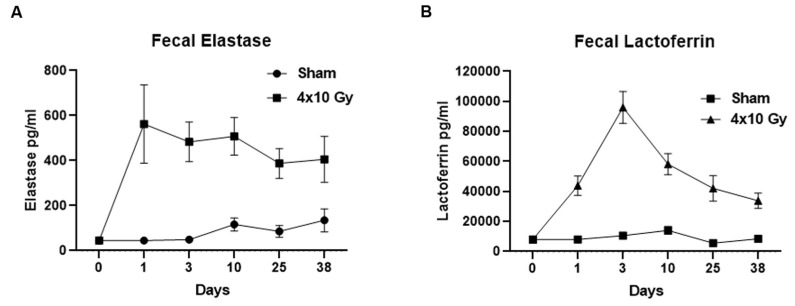
Quantitative analysis of intestinal inflammatory fecal markers elastase (**A**) and lactoferrin (**B**) 1, 3, 10, 25, and 38 days after X-ray exposure (4 × 10 Gy). RX vs. sham: elastase: 1, 3, and 10 days *p* < 0.0001, 25 and 38 days *p* < 0.01; lactoferrin: 1, 3, and 10 days *p* < 0.0001, 25 and 38 days *p* < 0.001. *n* = 12 each experimental group. Data are reported as mean ± SEM.

**Figure 7 ijms-24-08800-f007:**
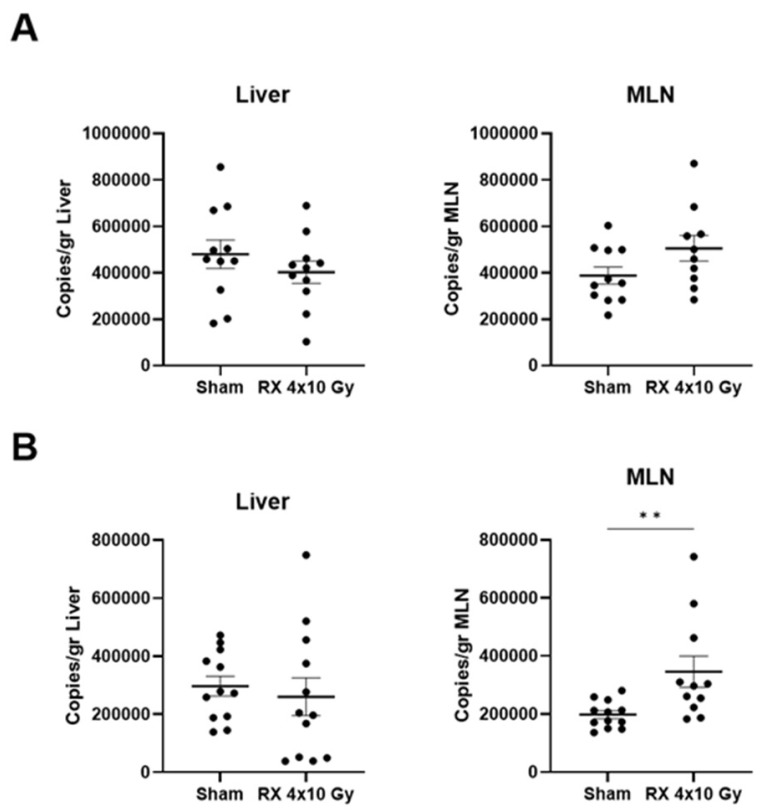
Quantification of bacterial translocation in liver and mesenteric lymph nodes 3 days (**A**) or 38 days (**B**) after X-ray exposure (4 × 10 Gy). Data are reported as mean ± SEM. ** *p* < 0.01.

**Figure 8 ijms-24-08800-f008:**
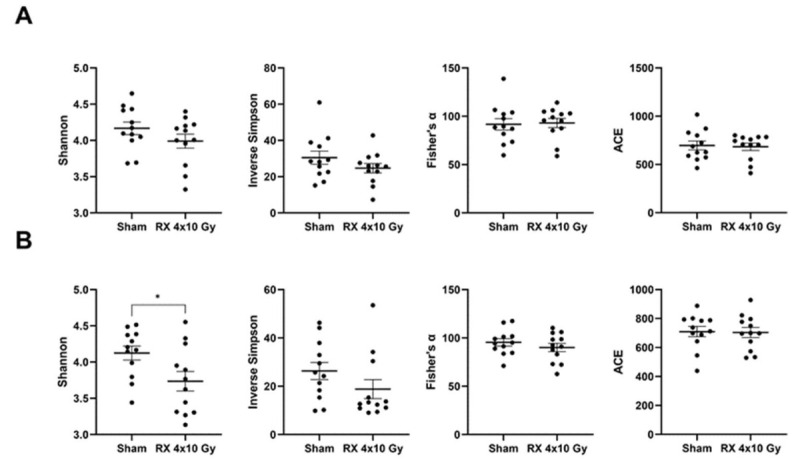
Quantification of alpha-diversity indices in mice microbiota at 3 days (**A**) or 38 days (**B**) after X-ray exposure (4 × 10 Gy). Data are reported as mean ± SEM. * *p* < 0.05.

**Figure 9 ijms-24-08800-f009:**
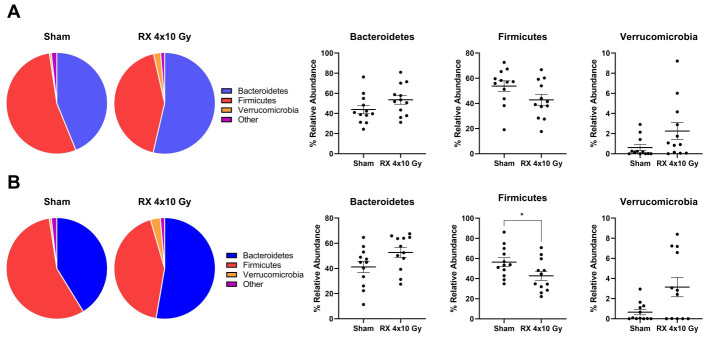
Pie chart and boxplot representation of dominant phyla in mice microbiota at 3 days (**A**) or 38 days (**B**) after X-ray exposure (4 × 10 Gy). Data are reported as mean ± SEM. * *p* < 0.05.

**Figure 10 ijms-24-08800-f010:**
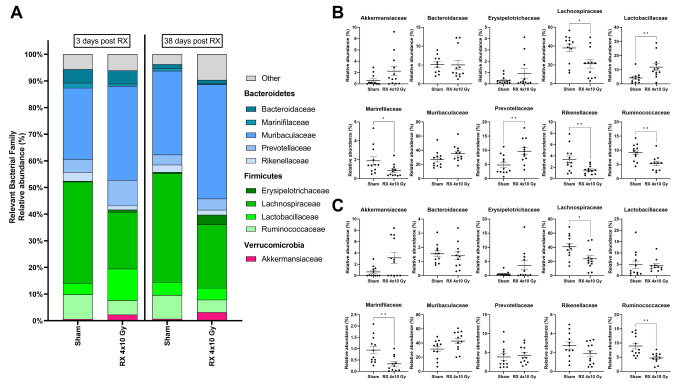
Merged bar plot (**A**) and boxplot representation of the 10 most represented families within the samples of mice at 3 days (**B**) or 38 days (**C**) after X-ray exposure (4 × 10 Gy). Data are reported as mean ± SEM. * *p* < 0.05, ** *p* < 0.01.

## Data Availability

Other datasets analyzed during the study are available from the corresponding authors on reasonable request.

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
