# Peer review of "Characterization of Early and Late Damage in a Mouse Model of Pelvic Radiation Disease"

_ijms, 2023, doi:10.3390/ijms24108800_

Round 1

Reviewer 1 Report

The manuscript by Vitali et al., is designed to develop a preclinical model for early and late pelvic radiation disease. The experiments were done carefully and the results were presented in a clear fashion.  However, there are few weaknesses in the study and some missed opportunities to generate novel data from the proposed model.  

1-    Investigation of DNA injury in the model was not proposed or attempted in earlier time points (e.g 1 hr following IR)?  Is there a reason for this?  Or the markers of IR- induced DNA damage was assessed and the results did not show significant changes?

2-    The model chosen was 4 x 10 Gy, which is at the SBRT dose range.  However when patients receive SBRT, usually 48 hrs between the fractions is preferred.  Here there were only 24 hr between fractions. Perhaps in future studies the investigators would consider this point to be more clinically relevant?

3-    mRNA expression of iNOS or COX2 should hardly be considered as markers of oxidative stress.  If the authors would like to claim, their preclinical model show increased oxidative stress at early time points, they need to be more rigorous for the markers they pick (preferably at protein level).  They could present IHCs of protein oxidation (3NT) or lipid oxidation (4HNE modified proteins) makers.  Or in tissue homogenates, they could measure markers for thiol oxidation (GSH, TrX etc).

4-    The use of quite young mice (and only males) is a weakness of the study.  I hope the investigators would further develop this model with mature adult mice from both sexes in the future.

5-    Did authors look at any tight junction proteins in the tissues (IHC) to also assess the degree of mucosal permeability? 

6-    I think there is a missed opportunity to use this model to look at microbiome changes in this preclinical model.  Can the authors comment on this in their discussion?

7- There is a typo in the Figure 6B.  Fecal lactoferrin was misspelled (as lattoferin)

Author Response

Reviewer 1

The manuscript by Vitali et al., is designed to develop a preclinical model for early and late pelvic radiation disease. The experiments were done carefully and the results were presented in a clear fashion.  However, there are few weaknesses in the study and some missed opportunities to generate novel data from the proposed model.  

1-    Investigation of DNA injury in the model was not proposed or attempted in earlier time points (e.g 1 hr following IR)?  Is there a reason for this?  Or the markers of IR- induced DNA damage was assessed and the results did not show significant changes?

Answer: In radiobiology, g-H2AX is classically used to evaluate radiation-induced DNA damage at early post-irradiation times (usually at 30 min post-irradiation). g-H2AX labels DSBs, but the quantification of the number of individual g-H2AX labelled foci in tissue sections is often problematic and g-H2AX is not only a marker of DNA damage, but it is also increased during cellular replication, present in the intestinal crypts.  To overcome this technical limitations, in our experimental design we opted for measuring apoptosis, that is triggered by DNA damage, and it is therefore an indicator of unrepaired DNA damage. This choice, besides the analysis of early time points (3 h), also allowed to carry out the evaluation of the number of apoptotic cells and BAX expression at later times, i.e. 3 days after the last X-ray exposure, when g -H2AX would have probably been undetectable.

2-    The model chosen was 4 x 10 Gy, which is at the SBRT dose range.  However when patients receive SBRT, usually 48 hrs between the fractions is preferred.  Here there were only 24 hr between fractions. Perhaps in future studies the investigators would consider this point to be more clinically relevant?

Answer: Thanks a lot for your comment. We fully agree that 40 Gy in 4 fractions represents a typical regimen of stereotactic radiotherapy, and that ultra-hypofractionated treatments are frequently delivered every other day, in clinical routine, in order to improve treatment tolerability. In fact, a prolongation of the duration of radiotherapy is associated with a reduction in the risk and severity of acute side effects. However, not all centers adopt this strategy. For example in the PACE-B randomized trial, where radiotherapy was used with doses similar to those of our experiment (PTV dose was 36.25 Gy in five fractions to the PTV and 40 Gy to the CTV over 1–2 weeks) the choice of the irradiation protocol (daily or alternate days) was left to center discretion. Anyway, in our study we chose the alternative of daily treatment, i.e. the theoretically more toxic one, precisely because we had the aim of studying the mechanisms of radio-induced toxicity on the bowel. 

3-    mRNA expression of iNOS or COX2 should hardly be considered as markers of oxidative stress.  If the authors would like to claim, their preclinical model show increased oxidative stress at early time points, they need to be more rigorous for the markers they pick (preferably at protein level).  They could present IHCs of protein oxidation (3NT) or lipid oxidation (4HNE modified proteins) makers.  Or in tissue homogenates, they could measure markers for thiol oxidation (GSH, TrX etc).

Answer: In this first study we characterized the PRD model at macroscopic, microscopic and molecular level. Our aim was not focused specifically on the oxidative stress, so we did not collect tissue homogenates for protein analysis. In our analysis we found that iNOS expression was increased only 3h after X-ray exposure and that 3 day after irradiation its expression returns to control values. To better investigate on this end-point, future studies could be carried out taking in consideration more suitable time points and sampling. Currently, we toned down our claim on the use of mRNA expression of iNOS or COX2 as oxidative stress markers.

4-    The use of quite young mice (and only males) is a weakness of the study.  I hope the investigators would further develop this model with mature adult mice from both sexes in the future.

Answer: We apologize to have not properly specified the age the animals had at the beginning of the experiments. The C57Bl/6 male mice were 49-55 days old when arrived in our animal facility. Treatments started after two weeks of acclimatation, when mice were 63-69 days (9-10 weeks) old. This information has been added to the Materials and Methods section of the revised manuscript. When 9-10 weeks old, C57Bl/6 mice are considered young adult animals and, according to the literature, they are widely used in PRD mouse models. Nevertheless, we agree that in future studies it would be interesting to investigate PRD induction also in older animals, considering the changes in inflammatory and immune responses occurring during ageing.  

We set up our PRD model on male mice according to the literature. Indeed, no differences in radiation-induced intestinal side effects were observed between the two sexes in pre-clinical research (Wang B, 2021). Moreover, on the technical point of view, our shielding systems could ensure protection of testis from exposure to radiation. At variance, protection of ovaries, and therefore induction of indirect hormonal driven effects, in female mice, while ensuring exposure of colon, would have been more critic. Nevertheless, as in PRD murine models, sexual dimorphism in gut microbiota seems to determine differences in the efficacy of therapeutic strategies for radiation toxicity at early time (Cui M, 2019), we are developing technical solutions to expose female mice in future studies. This aspect has been explained in the discussion of the revised version of the manuscript.

5-    Did authors look at any tight junction proteins in the tissues (IHC) to also assess the degree of mucosal permeability?

Answer: In the original version of the manuscript, the gene expression of the tight junction component (ZO-1) was analyzed in the colon tissue of mice sacrificed 3 hours after treatment with 4x10 Gy, by Real-Time PCR. As reported in Figure S1, no significant change was found. Consequently, we did not investigate this parameter by IHC in the first version of the manuscript.

However, to satisfy the request of this reviewer, we carried out an immunofluorescence for ZO-1 (3h and 3 days after X-Ray treatment) to better investigate tight junction functionality. Immunofluorescence was carried out on samples stored in paraffin during the study. Unfortunately the results showed high background and the classic reticulated coloration typical of ZO1 was not highlighted, but we found rather a diffused signal, making the data not quantifiable. We are in agreement with the Reviewer that this aspect is interesting but to be evaluated a dedicated sampling (OCT) should be considered. In our study this was not possible having only one cm of irradiated colon to process and we opted for sampling only for molecular analysis and standard histology.

6-    I think there is a missed opportunity to use this model to look at microbiome changes in this preclinical model.  Can the authors comment on this in their discussion?

Answer: Microbiome analysis was not included in the paper, because the data should be inserted in a possible future paper where our drugs are described. 

Following your valuable indication, now we add the data of microbiome profiling in the paper submitted.

7- There is a typo in the Figure 6B.  Fecal lactoferrin was misspelled (as lattoferin)

Answer: Typo error in Figure 6B has been corrected.

Reviewer 2 Report

Overall, this is a good body of work that characterized pathohistological changes in the colon after pelvic irradiation. Please see below a few comments to improve the quality and impact of the current manuscript.

1.     Please specify the size of the field for pelvic radiation therapy. It would be helpful to include a representative image. What method was used to measure the dose rate?

2.     Please justify why only males were used in the study.

3.     Please perform additional experiments to examine the vascular function and vascular density as well as tissue fibrosis at various timepoints post-IR

4.     Please stain tissue sections harvested at various time points post-IR to examine the number of Lgr5+ and Lrig1+ cells per crypt.

5.     Please examine the expression of marker genes for interferon signaling at various timepoints post-IR

Author Response

Reviewer 2

Overall, this is a good body of work that characterized pathohistological changes in the colon after pelvic irradiation. Please see below a few comments to improve the quality and impact of the current manuscript.

  1. Please specify the size of the field for pelvic radiation therapy. It would be helpful to include a representative image. What method was used to measure the dose rate?

Answer: The size of the field for pelvic radiation therapy was indicated clearly in the material and method section, and also in figure legend 1. However, to make the shielding system more intuitive, a representative image has been included in the revised version of the manuscript (Figure 1A).

Irradiation was delivered with a Gilardoni CHF 320 G Xray generator operated at 250 kVp, 15 mA, with HVL = 1.6 mm Cu (additional filtration of 2.0 mm Al and 0.5 mm Cu). At irradiation distance of 67.7 cm, the absorbed dose rate at the center of the irradiated volume in muscle was 0.89 Gy min-1 with relative expanded uncertainty of 10%, confidence level 95%. Dose monitoring was made using a PTW 7862 large-size plane parallel transmission chamber connected to a PTW IQ4 electrometer. The absorbed dose delivered at a given depth in an extended muscle phantom was determined from the measured value of the air kerma on the basis of the “in-air method”, and using the function "Percentage Depth Dose". Dose measurements were carried out by a cylindrical NE 2571 ionization chamber, coupled to a Farmer 2570/1 electrometer, calibrated in terms of Air Kerma at the Italian National Metrological Institute. This information is better explained in the revised version of the manuscript.

  1. Please justify why only males were used in the study.

Answer: According to the literature, we set up our PRD model on male mice. Indeed, no differences in radiation-induced intestinal side effects were observed between the two sexes in pre-clinical research (Wang B, 2021). In murine models of PRD, sexual dimorphism of gut microbiota seems to determine differences only in the efficacy of therapeutic strategies for radiation toxicity at early time (Cui M, 2019). Moreover, on the technical point of view, our shielding systems could ensure protection of testis from exposure to radiation. At variance, protection of ovaries, and therefore induction of indirect hormonal driven effects, in female mice, while ensuring exposure of colon, would have been more critic. However, as in PRD murine models, sexual dimorphism of gut microbiota seems to determine differences in the efficacy of therapeutic strategies for radiation toxicity at early time, we are developing technical solutions to expose female mice in future studies.

This aspect has been explained in the discussion of the revised version of the manuscript.

  1. Please perform additional experiments to examine the vascular function and vascular density as well as tissue fibrosis at various timepoints post-IR

Answer: The vascular function, vascular density and fibrosis are relevant end-point that can be analyzed to better define PRD; we thank the Reviewer to point out this aspect.

In order to examine vascular density and function, the number of vessels and their area (µm2) were analyzed on colon sections of sham- and X-ray irradiated animals, 38 days after the last irradiation. No significant variation in vascular area was found. Conversely, a significant increase in vascular density was found after X-ray treatment. We added this result in the revised version of the manuscript (Results section).

With regard to tissue fibrosis, the fibrotic score by histologic analysis as well as the expression analysis of genes involved in fibrotic process (i.e., alphaSMA and Col3a1) were carried out. We found no increase in the fibrotic score as well as in the expression of both analyzed genes.  Since fibrosis is a process that occurs following a long and extensive phase of unresolved inflammation, we think that 38 days after irradiation could be a time-point not sufficient to allow the establishment of a fibrotic state.

Since this aspect could be better analyzed in a dedicated study, with more appropriate time-points, the Authors preferred do not present these results in the revised version of the manuscript. However, graphs of the obtained results are reported below to the attention of this Reviewer.

  1. Please stain tissue sections harvested at various time points post-IR to examine the number of Lgr5+ and Lrig1+ cells per crypt.
  2. Please examine the expression of marker genes for interferon signaling at various timepoints post-IR

Answer to point 4 and 5: we understand the importance to evaluate the number of intestinal stem cells as well as to analyze the interferon signaling. Nevertheless, these aspects were not included in this study. Indeed, the aim of this study was to develop a PRD model, analyzing a wide variety of end-points both at early and late times. Further evaluations could certainly be addressed but we believe that these insights should be the subject of ad hoc designed future studies. On this issue, we trust this reviewer's understanding.

Round 2

Reviewer 2 Report

Revision accepted.